# Identification of Cancer Driver Genes by Integrating Multiomics Data with Graph Neural Networks

**DOI:** 10.3390/metabo13030339

**Published:** 2023-02-24

**Authors:** Hongzhi Song, Chaoyi Yin, Zhuopeng Li, Ke Feng, Yangkun Cao, Yujie Gu, Huiyan Sun

**Affiliations:** 1School of Artificial Intelligence, Jilin University, Changchun 130012, China; 2College of Computer Science and Technology, Jilin University, Changchun 130012, China

**Keywords:** graph neural network, multiomics data, cancer driver gene, PPI network, biomarker

## Abstract

Cancer is a heterogeneous disease that is driven by the accumulation of both genetic and nongenetic alterations, so integrating multiomics data and extracting effective information from them is expected to be an effective way to predict cancer driver genes. In this paper, we first generate comprehensive instructive features for each gene from genomic, epigenomic, transcriptomic levels together with protein–protein interaction (PPI)-networks-derived attributes and then propose a novel semisupervised deep graph learning framework GGraphSAGE to predict cancer driver genes according to the impact of the alterations on a biological system. When applied to eight tumor types, experimental results suggest that GGraphSAGE outperforms several state-of-the-art computational methods for driver genes identification. Moreover, it broadens our current understanding of cancer driver genes from multiomics level and identifies driver genes specific to the tumor type rather than pan-cancer. We expect GGraphSAGE to open new avenues in precision medicine and even further predict drivers for other complex diseases.

## 1. Introduction

Cancer is a complex disease with abnormal cellular metabolism. Cancer driver gene alterations always influence the normal-to-cancer cell transformation process and reprogram metabolism to meet the higher demands for nutrition and growth of uncontrolled cell proliferation [1]. Hence, it is critical to accurately identify such genes as the biomarkers for facilitating precision diagnosis and therapeutics [2]. Growing evidence suggests that cancer is driven by both genetic and nongenetic alterations [3], so integrating multiomics data and extracting valuable information from them may be an effective way to predict cancer driver genes.

Cancer progression is usually thought to result from the accumulation of driver genetic mutations which confer a selective growth advantage to the cell [2]. Many existing studies have been proposed to annotate cancer driver genes from genetic mutation data by counting mutation frequency, both alteration and conversion, and performing a series of statistical analyses, such as ccpwModel and xGeneModel [4]. In bladder cancer cells, the cancer driver genes reportedly tend to have a higher conversion frequency of C->G and C->T than other conversion patterns. Besides single nucleotide polymorphisms, a copy number variation (CNV) is also a major, well-studied mutation type. CanDrA [5] is a machine learning model for predicting cancer driver genes based on a set of 95 structural and evolutionary features. However, such methods only considering mutation frequency would overlook some real cancer driver genes with low mutation rates [6]. Moreover, aggregation mutation events in some genes on chromosomes lead to different mutation probabilities per base, so it is one-sided to identify cancer driver genes based on the average mutation rate [7]. On the other hand, although genome sequences have facilitated the identification of many cancer genes, the number of identified cancer genes specific to each tumor type is still low [8] and many genes that play important roles in tumorigenesis do not alter their DNA sequence [9].

Actually, cancer driver genes are usually dysregulated through various cellular mechanisms and signals, hence these nonmutated cancer-dependent genes, such as transcriptional and epigenetic regulators, are also of great interest. For example, DNA hypermethylation and hypomethylation at CpG islands surrounding gene promoters can inactivate tumor suppressor genes and activate oncogenes to promote tumor growth, respectively [10]. In addition to epigenetic effects, disrupting transcription factor binding sites also alters the regulation and expression of genes in several ways [11]. Therefore, it is necessary to integrate and make full use of the complementary information contained in multiomics data.

When looking at genes from the perspective of biological systems, they always work together. Existing studies have shown that cancer driver genes have the capacity to alter the gene expression of their interacting proteins or genes that share the same biochemical pathways [12], and the disruption of some driver genes may promote the progression of disease and lead to a cancer phenotype. Hence, taking advantage of the information contained in biological networks, such as PPI networks, is highly important for predicting cancer driver genes. Bashashati et al. proposed DriverNet [6] which identified driver mutated genes by analyzing their impact on the expression of genes connecting to them in the PPI network. The 20/20+ tool [9] generated a feature vector composed of mutation clustering, evolutionary conservation, functional impact of variants, type of mutation consequences, gene interaction network connectivity and other relevant factors for predicting cancer driver genes.

In combination with the valuable information contained in a graph and a deep learning model’s power, graph neural networks are widely used in classifying nodes in a network and perform well [13]. In recent years, some methods for classifying pan-oncogenes based on graph neural networks and multiomics data have been proposed. For example, Schulte-Sasse et al. designed the EMOGI [8] model, which combined CNV data, DNA methylation data, single nucleotide variant (SNV) data, and gene expression data and applied a graph convolution network with PPI networks to classify pan-cancer genes. In reality, due to the heterogeneity of tumors, cancer driver genes should be specific to each cancer type. However, EMOGI cannot distinguish the driver genes of cancer types so far. Furthermore, due to the limitation of graph convolution networks (GCNs) [14] in terms of efficiency, they are difficult to apply to large networks. Moreover, a GCN incorporates noise information in the way of aggregating nodes [15]. Compared with a GCN, GraphSAGE [16] aims to learn an aggregator rather than learning a feature representation for each node. Thus, the nodes and their neighborhood can be well-distinguished to reduce the influence of false-positive protein interactions in the network [17,18]. In each layer of a GraphSAGE model, the multiomics information of each gene and its interacted nodes, such as genes acting together with it in a signaling or regulatory pathway, as well as in protein complexes, are aggregated. In a real biological system, the strength of the interactions between the nodes usually varies, while a GCN assigns the same weight to each edge in the graph, leading to its limitations in practical application. A graph attention network (GAT) [19] incorporates an attention mechanism to assign weights to the edges between nodes for better learning the graph’s structural information and nodes’ representation.

In this study, we propose a novel framework called GGraphSAGE which integrates multiomics data, network-derived features, and graph neural networks for identifying cancer driver genes of each cancer type (Figure 1). We first generate a new feature vector for each gene in each tumor type, which is basically composed of four categories of features including 3 transcriptomic features, 1 epigenomic feature, 26 genomic features, and 6 network-derived features. Then, we take a total of 36 features along with the PPI network to train a graph neural network model. Considering a large number of genes cannot be definitely determined as cancer driver genes or not, we propose a semisupervised classification method to identify cancer driver genes, by applying one layer of a GAT and two layers of GraphSAGE to improve the performance of the model, where the GAT adds weights to each interaction of the PPI network and GraphSAGE is utilized to improve the robustness of the model and make it more suitable for semisupervised tasks [20]. In order to test the robustness of the model, we select TCGA-SKCM with a high mutation rate for validation. Experimental results demonstrate that for all eight selected tumor types, the classification performance of GGraphSAGE outperforms most existing cancer driver genes identification methods, and it also has a good potential for discovering newly predicted cancer driver genes (NPCDs).

## 2. Materials and Methods

### 2.1. Data Collection

#### 2.1.1. Multiomics Data

We collected RNA-seq data, SNV data, DNA methylation data, and CNV data of cancer samples and control samples from TCGA [21]. In our study, we used 3778 tumor samples and 258 control samples across eight tumor types for analyses as each of them had complete multiomics data (Table 1). Table 1 gives the detailed sample size of each tumor type.

#### 2.1.2. Network Data

We built the PPI network from the STRING database (https://cn.string-db.org/ accessed on 1 May 2021) and only considered the interaction whose score was greater than 0.9. We calculated the degree and betweenness centrality of each gene from the BioGrid network.

#### 2.1.3. Assignment of Positive Labels and Negative Labels for Genes

Aiming at classifying the genes, we first generated a positive or negative label for each gene. The positive label referred to high-confidence cancer driver genes of different tumor types from the intOGene library [22]. The negative label referred to genes that were most likely not related to cancer. To obtain a list of negative genes, we excluded the following five gene types from the known nonpositives and assigned negative labels to the remaining genes:Genes associated with the expression level of cancer driver genes.Genes related to existing cancer pathways.Cancer genes in the OMIM dataset [23] (https://omim.org//downloads accessed on 1 May 2021).Known cancer driver genes in the DriverDBV3 dataset (http://driverdb.tms.cmu.edu.tw/ accessed on 1 May 2021).Cancer driver genes in the NCG dataset [24] (http://ncg.kcl.ac.uk/ accessed on 1 May 2021).

Random pseudolabels were allocated to the above five types’ genes, and eventually, the model learned and predicted the labels by semisupervised learning. Table 2 shows the number of positive and negative labels in each tumor type.

### 2.2. Feature Generation

A gene’s features consist of its original multiomics information and network features derived from the PPI and BioGrid network. For each gene, for the transcriptome level, we computed the genes’ average expression and outlier ratio; for the genome level, we counted the mutation frequency, CNV, base transition frequency, variant allele fraction, and mutational heterogeneity signatures obtained from MutSigCV; for the epigenome level, we calculated the genes’ average methylation. For biomolecular networks, we combined the biological network and the genomic and transcriptomic data in a bipartite graph. Four bipartite graph features were extracted for each candidate gene (Figure 2). Two network features were calculated by the BioGrid network. The values of each feature of different scales were normalized by row before being joined into a feature matrix (Figure 3).

#### 2.2.1. Epigenetic Feature

For each gene *i* in tumor type $T_{j}$, we defined its epigenome feature as the fold change of mean methylation signal level (β) between the cancer and control samples:(1)$$Methylation_{T_{j},i}=\frac{{\bar{\beta }}_{T_{j,i}}^{C}}{{\bar{\beta }}_{T_{j,i}}^{N}}$$
where ${\bar{\beta }}_{T_{j,i}}^{C}$ and ${\bar{\beta }}_{T_{j,i}}^{N}$ represent the average methylation level of gene *i* in cancer and control samples of tumor type $T_{j}$, respectively.

#### 2.2.2. Transcriptome Features

##### Expression Fold Change

To quantify the differential expression level of each gene, we defined the log2 fold-change between the average expression level of cancer samples and the average expression level of normal samples as the expression fold-change feature as follows:(2)$$FoldChange_{T_{j},i}=\frac{{E\bar{X}P}_{T_{j,i}}^{C}}{{E\bar{X}P}_{T_{j,i}}^{N}}$$
where ${E\bar{X}P}_{T_{j,i}}^{C}$ and ${E\bar{X}P}_{T_{j,i}}^{N}$ represent the average expression of gene *i* in cancer and control samples of tumor type $T_{j}$, respectively. For the genes which were not measured in cancer or normal samples, we set them to 0.

##### Significance of Differential Expression

In addition to the average expression ratio, we conducted a Wilcoxon test and took the differences in the distribution of the gene expression between cancer and control samples as an attribute. If there was no significant difference, the attribute value of the gene was 0; if there was a significant difference and the mean value of the normal group was smaller than that of the cancer group, the attribute value of the gene was 1; otherwise, the attribute value was −1.

##### Gene Outlier Feature

The outlier feature of a gene was defined based on the Grubbs test. First, we generated a patient-outlier matrix to represent the specificity of these genes. For each gene *i*, we used the difference between the expression value of each sample and the mean of the expression values of all samples to obtain $G_{ij}$, which was calculated as follows:(3)$$G_{ij}=\frac{EXP_{ij}-mean\left(EXP_{i}\right)}{S_{i}}$$
where $EXP_{ij}$ is the expression level of gene *i* in tumor sample *j*, and $S_{i}$ is the standard deviation of the expression level of gene *i* in all samples.

Gp(N) was defined as follows:(4)$$G_{p}\left(N\right)=\frac{N-1}{\sqrt{N}}\sqrt{\frac{t_{\frac{\alpha }{2N},N-2}^{2}}{N-2+t_{\frac{\alpha }{2N},N-2}^{2}}}$$
where *N* represents the number of measurements, and $t_{\frac{\alpha }{2N},N-2}^{2}$ denotes the critical value of the *t*-distribution. α is the significance level and we set it to 0.05. α/2N is the significance level of the *t*-distribution with degrees of freedom for N−2. If there existed $G_{ij}>G_{p}\left(N\right)$, then the value of the *i*th row and *j*th column in the outlier matrix was 1, otherwise it was 0. Then, we calculated the sum of the outliers of all cancer samples corresponding to each gene from the patient-outlier matrix as the outlier feature for the gene.

#### 2.2.3. Genomic Features

##### Gene Mutation Frequency

For each tumor type, we generated a mutation matrix consisting of “1” and “0” with rows as genes and columns as samples. A “1” indicated that the gene had a single nucleotide variation in the sample, otherwise it was “0”. We computed the mutation frequency of each gene in all cancer samples.
(5)$$MF_{T_{j},i}=\frac{m_{T_{j},i}}{n_{T_{j},i}}$$
where $MF_{T_{j},i}$ represents the mutation frequency of gene *i* in the tumor type $T_{j}$’s samples, $m_{T_{j},i}$ and $n_{T_{j},i}$ denote the number of samples with mutations in gene *i* and the number of total samples of cancer type *j*, respectively.

##### Variant Allele Fraction

To analyze the tumor heterogeneity and tumor purity, we generated the variant allele fraction feature by the ratio of the coverage depth of mutant loci in tumor samples to the coverage depth of sequencing as follows: (6)$$Variant\;allele\;fraction=\frac{Allele\;Depth}{Total\;Depth}$$
where AlleleDepth denotes the depth of coverage of reads supporting mutant loci, and TotalDepth denotes the depth of coverage of the total reads of that locus.

##### MutSigCV Derived Attribute

Mutation rate is an important indicator for identifying cancer driver genes. However, due to the difference in length and CG base content between genes, the average mutation frequency across samples did not completely reflect the real mutation level. To avoid this situation, we selected 11 quantified features from MutSigCV [9], aiming to reveal the regional heterogeneity present in nucleotide mutations and to evaluate the impact of these features on the identification of cancer driver genes.

DNA replication time;Noncoding mutation rate;Local GC content;HiC compartment;Local gene density;Wgs mean depth;Wgs percent 20x;Capture on target rate;Capture mean depth;Capture pct200;Capture mean percentGC.

##### Mutant Base Conversion

Existing studies have shown that there are significant differences in the frequency of base interchange in cancer. For example, the frequency of the CpG dinucleotide transition mutation in gastrointestinal cancer (esophagus, colon, rectum, and stomach) is high [25]. The frequency of the C->A and C->G type mutagenesis in bladder cancer cells is higher than that in other types of tumor cells [9]. Herein, we counted the number of base conversion corresponding to each gene in each sample and generated 12 features, namely C->A, C->T, C->G, A->T, A->C, A->G, G->C, G->A, G->T, T->A, T->G, and T->C.

##### Copy Number Variation Rate

We defined the copy number variation rate of a particular gene *i* in tumor type $T_{j}$ as:(7)$$CNV_{T_{j},i}=\frac{\sum_{k=1}^{m}tumorS_{i,T_{j}}^{k}\times n}{\sum_{l=1}^{n}normalS_{i,T_{j}}^{l}\times m}$$
where $tumorS_{i,T_{j}}^{k}$ and $normalS_{i,T_{j}}^{l}$ represent the number of amplifications or deletions of gene *i* in cancer sample *k* and normal sample *l* from the tumor type $T_{j}$, respectively. *m* and *n* are the number of cancer and normal samples.

#### 2.2.4. Biological Network Derived Features

##### Bipartite Graph Attributes

For each tumor type, we used the gene expression matrix, mutation matrix and PPI network to construct a bipartite graph (Figure 2). Nodes in the left partition of the bipartite graph corresponded to mutated genes, and nodes in the right partition represented the outlier expression status of each patient. We drew an edge between two nodes in different partitions under the following conditions: in the same patient, mutations of the left-partition node gene *i* and the abnormal expression of the right-partition node gene *j* were found, and there was an interaction between gene *i* and gene *j* in the biological network. To estimate the effect of each mutated gene on DNA replication, we added cell growth attribute to each gene according to whether it belonged to cell-proliferation-related biological functions or pathways in the bipartite graph and gave four features to each mutated gene: the number of connected edges, the number of edges connecting to cell-proliferation-related genes (CPRG), the gene’s cell growth function attribute, and the number of patients covered by the gene (Figure 2).

##### BioGrid Network Features

We separately calculated the degree and betweenness centrality of the genes based on the BioGrid network. The degree centrality score is the number of connected edges of each gene in the network; the betweenness centrality is the ratio of the shortest paths in the network that pass through a gene and connect two genes to the total number of shortest path lines between these two genes. These two features reflected the importance and the topology characteristics of each gene in the network.

### 2.3. Model Construction

The proposed semisupervised deep graph learning framework GGraphSAGE was constructed using a GAT layer and two GraphSAGE layers, where it took the PPI network as the graph and each gene on the PPI network as a node. The advantage of GGraphSAGE is that it aggregates the neighborhood information in the graph more reasonably. The GAT quantifies the differences in interactions between homologous partners in biological large-scale networks by aggregating the weights of adjacent edges. At the same time, by sampling from the inside out and aggregating from the outside of GraphSAGE, using “concat” instead of “average”, it can better distinguish the information of itself from its neighbors. We used the generated feature matrix and PPI networks as inputs, which allowed GGraphSAGE to receive both multiomics information and molecular interactions in biological systems. It also greatly helped to improve the overall performance of the model (Figure 4).

#### 2.3.1. GraphSAGE Layer

GraphSAGE aims to improve the efficiency of a GCN and reduce noise. It learns an aggregator rather than the representation of each node, which enables one to accurately distinguish a node from its neighborhood information. In addition, it can be trained in batches to improve the polymerization speed. Compared with a GCN, GraphSAGE was more flexible and suitable for our semisupervised task. The GraphSAGE algorithm is formulated as follows:(8)$$h_{v}^{0}\leftarrow x_{v},\forall v\in V$$
(9)$$h_{N(v)}^{k}\leftarrow AGGREGATE_{k}\left(\left\{h_{u}^{k-1},\forall u\in N\left(v\right)\right\}\right)$$
(10)$$h_{v}^{k}\leftarrow \sigma \left(W^{K}\cdot CONCAT\left(h_{u}^{k-1},h_{N(v)}^{k}\right)\right)$$
where $x_{v}$ is the input feature, *k* is the depth of the feature matrix, $h_{N(v)}^{k}$ is defined as the aggregated representation of the neighbors of node *v*, $h_{v}^{k}$ denotes the representation of the *v* node after aggregating its neighbors, and *W* is a learnable parameter.

#### 2.3.2. GAT Layer

Since there are significant differences in the level of signal transduction between genes in biological systems, it is reasonable to set different weights for each edge in the PPI network. A GAT can set weights to each edge in the graph and was used to distinguish edges with various weights in the PPI network. A GAT computes the weight of each edge as follows:(11)$$a_{ij}=\frac{exp(LeakyReLU\left(a^{T}\left[Wh_{i}\|Wh_{j}\right]\right))}{\sum_{k\in N_{i}}^{}exp(LeakyReLU\left(a^{T}\left[Wh_{i}\|Wh_{k}\right]\right)}$$
where $a_{ij}$ represents the weight relation coefficients of node *i* and node *j*, $a^{T}$ is a learnable vector ($a^{T}\in R^{1\times n}$), and *n* is feature dimension of each node. *W* is a learnable weight matrix ($W\in R^{n\times n}$). $h_{i}$ denotes the embedding of node *i*. $N_{i}$ are all the neighbor nodes of *i*.
(12)$$h_{i}^{{}^{\prime }}=\sigma \left(\sum_{j\in N_{i}}^{}a_{ij}Whj\right)$$
where σ is defined as the nonlinear activation function. $h_{i}^{{}^{\prime }}$ denotes the embedding of node *i* after aggregating all the neighboring nodes.

### 2.4. Model Training

We randomly divided all the samples into a training set and test set with the proportion of 70% and 30%, respectively, and took the generated feature matrix with partial node labels and the PPI network as inputs. The cross-entropy loss *L* for our training nodes was defined as:(13)$$L=\frac{1}{N}\sum_{g}^{}L_{g}=\frac{1}{N}\sum_{g}^{}-\left[y_{g}\cdot log\left(p_{g}\right)+\left(1-y_{g}\right)\cdot log\left(1-p_{g}\right)\right]$$
where $y_{g}$ is the label of gene *g*, with 1 representing the positive class and 0 representing the negative class. $p_{g}$ is the probability of gene *g* being predicted as positive. We implemented our framework using Torch [26] and optimized the training process with the Adam optimizer. The parameters for each layer are listed in Table 3.

## 3. Results

### 3.1. Evaluation Metrics

We evaluated the performance of the model across eight tumor types and compared it with eight methods for identifying cancer driver genes, including state-of-the-art methods and classic machine learning methods. We calculated the average precision (AP) for each tumor type, which is the area under the precision–recall curve (P-R curve) for each method to quantify the performance of the models. We drew the P-R curve according to the following:(14)$$Recall=\frac{TP}{TP+FN}$$
(15)$$Precision=\frac{TP}{TP+FP}$$
where TP is the number of genes correctly classified as positive by the model; FN denotes the number of genes incorrectly classified as negative by the model; and FP represents the number of genes incorrectly classified as positive by the model. We obtained the approximate area (AP) under the P-R curve by integration:(16)$$AP=\int_{0}^{1}p\left(r\right)dr$$

Because of the discreteness of the data, we actually smoothed the P-R curve rather than compute it directly. For each point on the P-R curve, the precision value took the value of the maximum precision to the right of that point. Next, on the smoothed P-R curve, the precision value of the 10 equipoints of the recall axis (including 11 points and breakpoints) was taken, and its average value was calculated as the final AP value. We calculated AP as follows:(17)$$P_{smooth}\left(r\right)=max_{r^{{}^{\prime }}>=r}P\left(r\right)$$
(18)$$AP=\frac{1}{11}\sum_{0,0.1\ldots 1.0}^{}P_{smooth}\left(i\right)$$
where $P_{smooth}\left(r\right)$ represents the precision value of the model when the recall value is *r* after the smoothing process. The AP is between zero and one, and the larger the AP, the better the performance (Figure 5).

### 3.2. Performance of GGraphSAGE by Comparing with SOTA Methods

To evaluate the performance of GGraphSAGE, we first compared it with three SOTA cancer driver gene identification methods, CanDrA [5], 20/20+ [27], and EMOGI [8]. CanDrA takes chromosome number, genomic coordinate, reference allele, mutated allele, and the strand of the mutation as the features of each gene and obtains a score to determine whether this gene is a driver or not. For 20/20+ [27], we used the mutation data from TCGA and 24 attributes acquired from BioGrid [28], MutsigCV [9], and SNVBox [29] as the features and applied a random forest [30] to predict the probability of the genes being drivers. For EMOGI [8], we used DNA methylation, RNA-seq, and CNV data collected from TCGA for different tumor types. We obtained the PPI network from the STRING database and only retained the interaction scores greater than 0.9. We randomized the labeled dataset to the training set (75%) and the test set (25%) for training and used a 10-fold cross-validation for the parameter optimization and improvement of the model stability.

The GGraphSAGE model performed better than all three SOTA methods. Overall, 20/20+ had stable performance across different cancer types but generally had a poorer performance compared with GGraphSAGE. CanDrA achieved good performance on STAD, LUAD, and LIHC, but it was overly dependent on the dataset and had unstable performance on the other five tumor types. EMOGI was a similar approach to GGraphSAGE in model structure and also used multiomics data and graph convolution networks. Although EMOGI was proposed for the identification of pan-cancer genes, it could also work in predicting cancer driver genes of specific tumor type. The performance of EMOGI was not good on LIHC and LUAD, while it performed well on other tumor types (Figure 5).

### 3.3. Performance of GGraphSAGE by Comparing with Classical Machine Learning Models

We also compared GGraphSAGE with three classical machine learning models, including K-nearest neighbors (KNN) [31], support vector machines (SVM) [32], and random forests.

KNN is a classical algorithm for supervised learning classification based on the distance between the node and the nearest k nodes and performs well in binary classification tasks. An SVM is a binary classification model. It is the nonlinear classifier defined on the feature space with maximum interval. A random forest is a classifier composed of many decision trees, in which each tree selects the optimal feature recursively and divides the training data according to the feature.

The results showed that the three models generally performed poorly overall on the tumor types, except for the SVM on LUAD and BLCA, and the random forest on LUSC. The AP of GGraphSAGE was generally 20% higher than the three machine learning models on all tumor types, which indicated that the addition of biological network structure information was helpful for cancer driver gene identification (Figure 5).

### 3.4. Ablation Experiments

To prove the superiority of the combination of a GAT and GraphSAGE, we calculated the AP of the GAT and GraphSAGE, respectively. We set the same inputs as for GGraphSAGE and only changed the model parameters. It can be seen that the performance of the GAT and GraphSAGE was lower than that of GGraphSAGE.

In the GAT, nodes in the graph can be assigned different weights based on the characteristics of their neighbors. The GAT is suitable for calculating the specificity interactions in PPI networks. However, there is noise (false-positive interacting protein) in the topology of the PPI network [33] which affects the performance of the GAT. GraphSAGE introduces neighbor sampling. Through the “concat” method of aggregating neighborhood nodes, GraphSAGE can distinguish itself from neighborhood information to reduce the influence of noise data and thus improve the robustness of the model [17,18]. Although GraphSAGE samples neighborhood nodes to improve the efficiency of training, some neighborhood information is lost. The method of node aggregation in GGraphSAGE improves the robustness of the model, allowing sampling nodes to be aggregated with nonequal weights, while preserving the integrity of the first-order neighborhood structure information.

As a result, GraphSAGE performed well on LIHC, ESCA, BLCA, and LUSC but moderately on the other four tumor types. The GAT performed well on STAD, ESCA, SKCM, and BLCA but poorly on the other tumor types, which indicated the performance of the GAT model was highly dependent on the data set and had a low stability. In contrast, GGraphSAGE performed well and was stable across all tumor types, which suggested that GGraphSAGE was superior to GAT or GraphSAGE alone.

Moreover, to evaluate the contribution of each type of data, we conducted ablation experiments on each type of data. Specifically, we removed one type of data in each experiment and used the remaining data to identify the cancer driver genes by GGraphSAGE. Then, we calculated the AP to assess the performance. The results showed that across the eight cancer types, the complete multiomics data (containing genomic, epigenetic, transcriptome, and network-derived features) performed best, which illustrated the necessity of leveraging the complete multiomics and network data (Figure 6). After removing any one type of features, the performance of the model decreased, which proved that each type of data contributed to the model. For eight cancer types, removing genomic data had the biggest effect on model performance (a mean 39.3% decline in AP), while removing epigenetic data had the least effect (a mean 10.7% decline in AP). It indicated that genomic features contributed the most to the classification and prediction of cancer driver genes, while epigenetic features contributed the least. Compared to other cancer types, the performance was poor after removing transcriptome features in BLCA, LUAD, and STAD, which indicated that transcriptome features contributed significantly to these three cancer types, while network-derived features contributed more to the other types.

### 3.5. Association between Newly Predicted Genes and Cell Proliferation

Uncontrolled cell proliferation is a determinant of carcinogenesis and driver mutation. To assess the effect of mutations on cell proliferation, we analyzed the differences in abnormal proliferation levels between samples with and without mutations (Figure 7). Specifically, we first retrieved 160 CPRGs from GSEA that were included in DNA replication and cell cycle pathways, generated the transcriptomics data matrix of N samples on the CPRG, and calculated the Spearman correlation coefficient between these genes by setting 0.4 as the threshold. For the active cancer-related biological process, most genes worked together and more than half of genes were significantly coexpressed, so we set the low significance cutoff of correlations as $10^{-3}$ and selected a set of top 20 genes which were highly correlated with each other as core genes to represent the whole cell proliferation process. Subsequently, we used the linear regression coefficient between core genes’ expression value of each sample and core genes’ average expression value vector on N samples, as the cell proliferation activity of each sample. The objective function of the linear regression was:(19)$${\hat{S}}_{\theta }=\theta \cdot S$$
(20)$$RST=\sum_{i=1}^{n}{\left(S_{mean}^{\left(i\right)}-{\hat{S}}_{\theta }^{\left(i\right)}\right)}^{2}$$
where θ represents the weights of samples, that is, the parameter that minimizes the squared term of the residual; *S* is the sample column ($S\in R^{1\times 20}$) in the core gene matrix; ${\hat{S}}_{\theta }$ represents the predicted value of a linear regression function that is used to fit the mean vector of the sample (${\hat{S}}_{\theta }\in R^{1\times 20}$); and $S_{mean}$ denotes the mean expression level of core genes across all samples. We took θ as the samples’ cell proliferation level.

To evaluate the effect of a mutation on cell proliferation, we divided the cancer samples into two groups: one with such mutation and the other without (M and U in Figure 7). Then, we conducted the Wilcoxon test analysis to assess the significance of differences in cell proliferation between these two groups. We expected the driver mutations to cause abnormal proliferation, but the passenger mutations were not associated with abnormal proliferation. We identified the top 20 candidate genes for each tumor type according to the scores (the score was the probability of being predicted as cancer driver genes in the model) obtained from GGraphSAGE and removed genes that overlapped with those of the intOGene database. Candidate genes with a *p*-value less than 0.05 indicated a significant alteration of the corresponding gene was closely related to cell proliferation and thus were ultimately selected as cancer driver genes to further demonstrate their association with cancer (Figure 8). The information of the candidate NPCDs is listed in Table 4.

### 3.6. Newly Predicted Cancer Driver Genes and Their Verified Functions

On the basis of the cell proliferation analysis, we further verified the functions of the candidate driver genes. Most of the genes have been extensively studied and are directly and indirectly associated with tumor development in the published literature. Some genes are identified as molecular markers of carcinogenesis. For example, CUL1 is defined as a biomarker for breast cancer because it significantly promotes breast cancer cell migration, invasion, and test-tube formation, as well as angiogenesis and metastasis in vivo [34]. Alternatively, the expression levels of some genes affect the functions involved in the transformation of normal cells into cancer cells. For example, ACTN2 overexpression in human hepatoma cells shows an enhanced cell viability and invasion, suggesting that it may play a role in late metastasis, such as exosmosis and lung colonization [35]. Besides extensive studies on gene expression level and function, some genes influence the occurrence and development of cancer at the genomic and epigenetic levels. Sucularli et al. reported that the deleterious mutations of DYNC1H1 led to the formation of associated cancers [36]. Lin et al. found the methylation of RILP in lung cancer promoted tumor cell proliferation and invasion [37]. Hence, we conclude that the NPCDs are expected to provide guidance for researchers on further studies.

Moreover, eight genes predicted by GGraphSAGE were also coauthenticated by the most popular 30 SOTA methods (e.g., CHASM [29], e-Driver3D [38], OncoIMPACT [39], PolyPhen2 [40], CTAT-score [41]). These genes include TBX3, CUL1, and MAP2K4 for BRCA, PTCH1 for ESCA, ASXL2 for BLCA, EZH2 for LIHC, RIT1 for LUAD, and ERBB4 for STAD.

In summary, we demonstrated that GGraphSAGE had a higher performance than other state-of-the-art methods based only on multiomics, biological networks, or using simple graph neural networks in the classification of cancer driver genes by tumor type. We also provided strong evidence for the credibility of new driver genes predicted by GGraphSAGE at both theoretical and function levels through a literature review and functional correlation analysis (Table A1).

## 4. Discussion

Carcinogenesis is usually thought to result from the accumulation of key alterations, both at the genetic and nongenetic level. Hence, the identification of cancer driver genes is important to discover drug targets for specific cancer types. In this study, we proposed a novel semisupervised graph-neural-network-based framework, GGraphSAGE, which integrated multiomics and network-derived features to identify cancer driver genes for each cancer type. We evaluated the performance of GGraphSAGE against three state-of-the-art methods and three classic machine learning methods under the average precision (AP) index across eight tumor types. Experiments showed that GGraphSAGE performed better than all these methods in both precision and stability. Considering cancer is a highly heterogeneous and complex disease, the disease initiation mechanism, microenvironment composition, and key alterations of various cancer types should be intuitively distinct [42]. Hence, different from most existing methods focusing on pan-cancer driver genes, we proposed to identify cancer driver genes based on the characteristics specific to each cancer from various aspects. Besides widely studied genomic features, we also focused on nongenetic alterations including transcriptomic and epigenomic features, as well as biological-network-derived features, and generated a feature vector of length 36 for each gene at the end. It is well known that genes in the biological systems do not work independently, and the disruption of some driver genes may promote the initiation and progression of cancer. Hence, taking advantage of the information contained in biological networks is highly important for predicting cancer driver genes. Indeed, most advanced methods also take this into account. The combination of valuable information contained in graph and deep learning approaches, graph neural networks have gradually been used in classifying nodes with outstanding performance [8,13]. Herein, considering large numbers of genes cannot be definitely determined as cancer driver genes or not, we proposed a semisupervised classification method to identify cancer driver genes, applying a GAT and GraphSAGE to improve the performance of the model. Ablation experiments demonstrated that the performance of jointly applying the GAT and GraphSAGE models was much better than using either one alone. Moreover, the application of GraphSAGE was able to provide an availability assurance for scalable networks. After years of efforts, some public databases, such as COSMIC [43] and intOGene, have release parts of driver genes. Beyond these, GGraphSAGE was able to identify new driver genes with a high potential. Through a literature review and functional correlation analysis (**see Results**), we demonstrated the credibility of the newly predicted ones. We took abnormal cell proliferation as an indicator of carcinogenesis and determined whether a candidate gene was a driver by evaluating the association between its alterations and this indicator. In addition, other key biological functions, such as EMT [44], which is an important cancer development indicator, will also be considered in our further studies for identifying key drivers of cancer progression.

The application area of the GGraphSAGE model is broad, as it can be applied to any multiomics data and biological network other than this study. Moreover, the predicted diseases are not limited to cancer but can also be applied to other complex diseases. Our GGraphSAGE is designed to classify cancer driver genes and passenger genes in various tumor types. The semisupervised mechanism of the model also identifies NPCDs that play a critical role in tumor development and the impact on cell growth function (Table A1). These NPCDs are highly similar to the cancer driver genes in terms of multiomics embedding and biological network structure. The GGraphSAGE framework provides an important analytical tool for the future of precision medicine and for understanding the process of tumor development and targeted therapy.

## 5. Conclusions

GGraphSAGE is a graph neural network framework for accurately and efficiently distinguishing cancer driver genes, which is empowered by the generated comprehensive multiomics features and the combination of GraphSAGE and GAT models. Through statistical analyses, we found that the alterations of some newly identified cancer driver genes influenced cell proliferation function and were reportedly associated with cancer initiation and development. The GGraphSAGE framework provides a new insight to identify the driver genes of complex disease and is helpful to understand the process of disease development and design targeted therapy.

## Figures and Tables

**Figure 1 metabolites-13-00339-f001:**
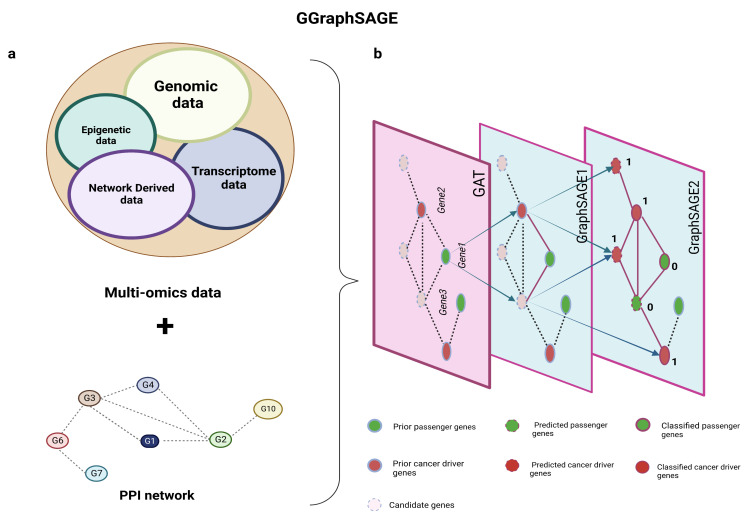
**Schematic diagram of GGraphSAGE framework.** (**a**) Inputs of GGraphSAGE framework include multiomics data (transcriptomic, genomic, and epigenetic data) and network and network-derived data; (**b**) all features and the PPI network are used for the semisupervised training of cancer driver genes. During the training of the GGraphSAGE model, the features are compressed by a three-layer graph aggregate operation and gradually include an increasingly wide range of neighboring nodes. The final output is determined by a probabilistic model (of whether it is a cancer driver gene).

**Figure 2 metabolites-13-00339-f002:**
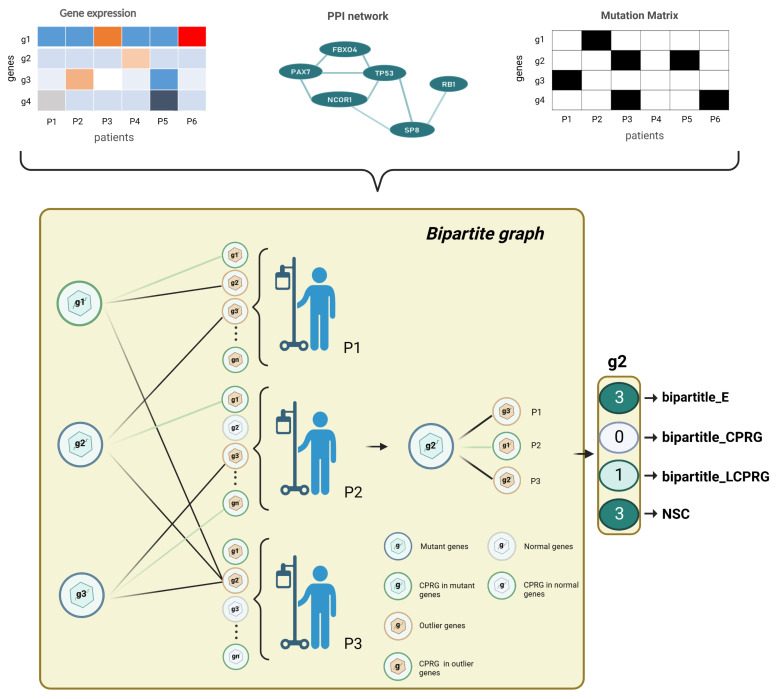
**Bipartite graph architecture.** For each tumor type, we built a bipartite graph by gene expression data, mutation data, and PPI network. We finally output a 4 × n dimensional feature matrix (n is the number of genes to be predicted). The left partition of the bipartite graph is the mutated genes, and the right partition of the bipartite graph is the outlying expression status for each tumor sample. bipartite_E represents the number of edges of a mutated gene; bipartite_LCPRG denotes the number of edges of a mutated gene linked to CPRGs; bipartite_CPRG is a binary number: if a gene was a CPRG, we set the bipartite_CPRG feature of the gene as 1, otherwise 0; NSC is the number of patients covered by a mutated gene.

**Figure 3 metabolites-13-00339-f003:**
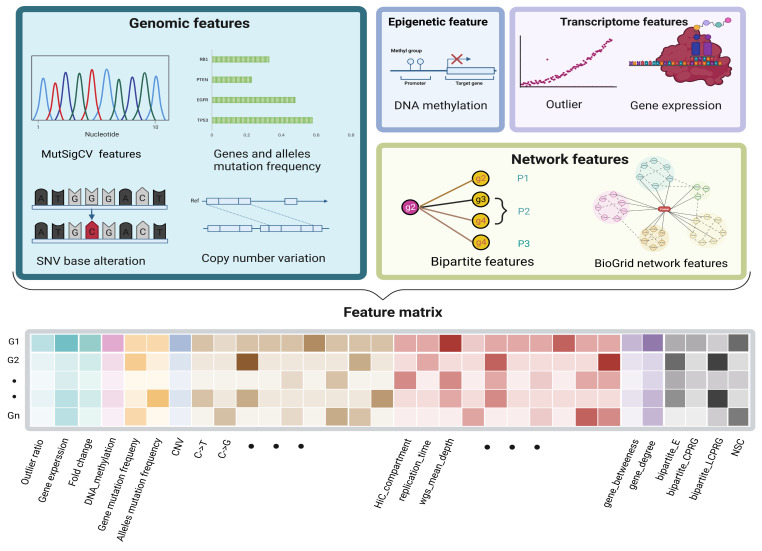
**The characteristics of each gene.** The attributes of a gene included the following four parts, 26 genomic features, 3 transcriptomic features, 1 epigenomic feature, and 6 network derived features from the bipartite graph and BioGrid network, which together formed a 1× 36-dimension feature vector for each gene.

**Figure 4 metabolites-13-00339-f004:**
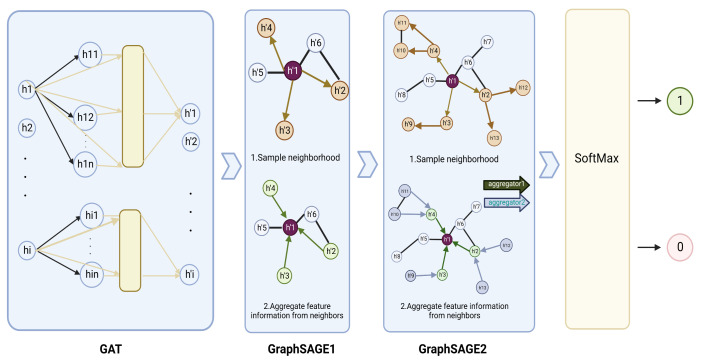
**Graph neural network structure of GGraphSAGE framework.** In the GGraphSAGE model, the feature matrix is aggregated around nodes through a GAT layer and two GraphSAGE layers for aggregating node, and each node contains the 3rd-order neighborhood information of the node. The final output is that each node (gene) is assigned a two-dimensional vector by the softmax layer, which consists of the probability that the node is a driver cancer gene and the probability that the node is a passenger gene. A label (1 or 0) is set to each node by comparing the magnitude of the two probabilities.

**Figure 5 metabolites-13-00339-f005:**
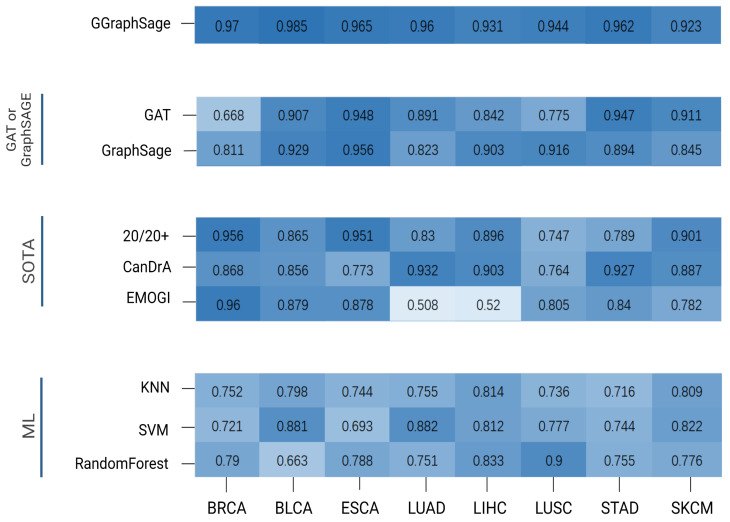
**Performance comparison of GGraphSAGE with other methods.** The heatmap reveals the performance (AP) comparison of different methods for each tumor type, with darker colors indicating higher AP values. These methods were divided into 4 categories: GGraphSAGE: the combination of GAT and GraphSAGE; GAT or GraphSAGE: GAT or GraphSAGE model only; SOTA methods: 20/20+, CanDrA, and EMOGI; ML (machine learning): KNN, SVM, and random forest. As can be seen from the figure, GGraphSAGE has a high AP value on each tumor type, and its performance is better than other methods.

**Figure 6 metabolites-13-00339-f006:**
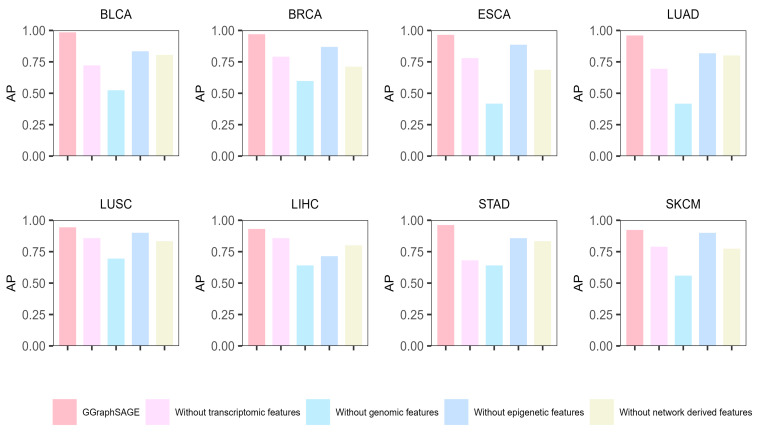
**Results of ablation experiment of GGraphSAGE in eight cancer types.** The bar chart represents the model performance (AP) of GGraphSAGE across eight cancer types when removing genomic, epigenetic, transcriptome, and network data, respectively.

**Figure 7 metabolites-13-00339-f007:**
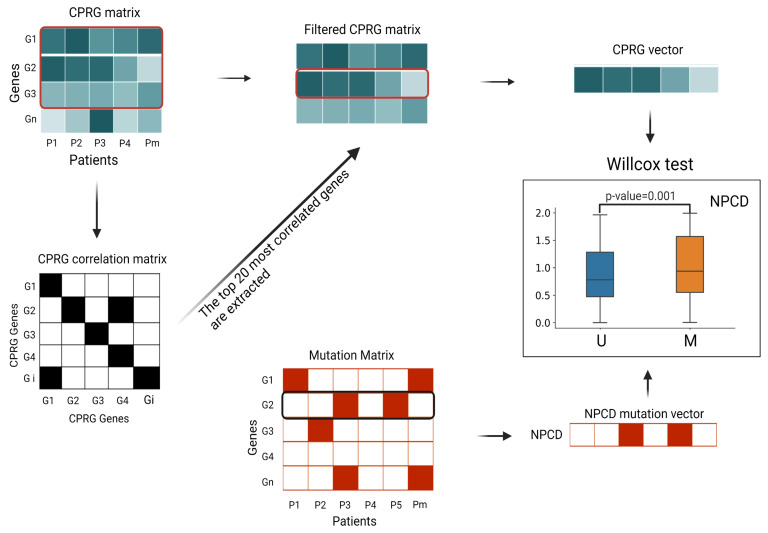
**Flow chart of DNA replication function correlation test.** (1) We extracted the fraction of all CPRGs from the $gene\;expression\;matrix\in R^{20500\times n}$ (n is the number of samples) of each tumor type and created the $CPRG\;matrix\in R^{160\times n}$. (2) According to the Spearman correlation coefficient threshold, we generated $CPRG\;correlation\;Matrix\in R^{160\times 160}$(symmetric matrix). (3) We selected the top 20 genes with the highest correlation in the CPRG matrix and extracted the top 20 genes from the CPRG matrix to produce the $Filtered\;CPRG\;matrix\in R^{20\times n}$. (4) The mean expression level of each CPRG in the filtered CPRG matrix was used to calculate the linear regression parameters for each sample, as the CPRG vector for the sample. The CPRG vectors were divided into mutation/nonmutation vectors according to the mutation matrix, and the Wilcoxon test was performed on the distributions of these two vectors to compute the *p*-values. The box plots are drawn for each NPCD by the two sample-population distributions (Figure 8).

**Figure 8 metabolites-13-00339-f008:**
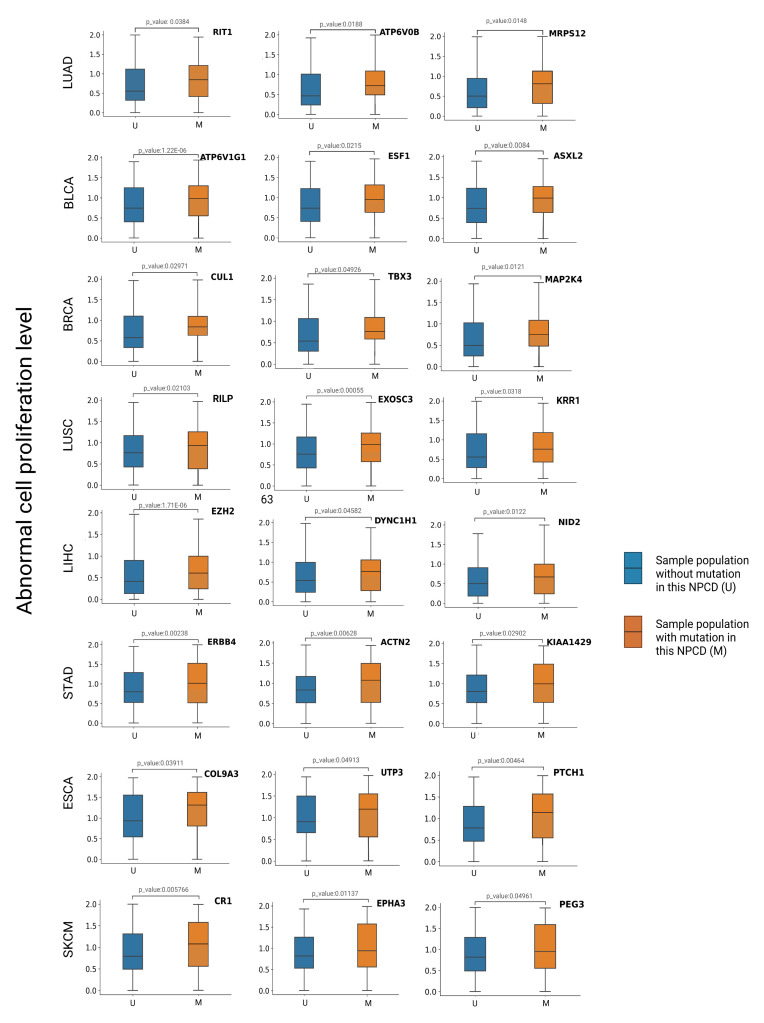
**Box plots of abnormal cell proliferation level of CPRG.** The box plots show the differences in the distribution of abnormal proliferation levels between samples with and without mutations in different tumor types. The blue and orange boxes represent the distribution of the U and M sample populations, respectively. The vertical axis is the abnormal cell proliferation level of each sample. We conclude that mutations in candidate genes with a *p*-value < 0.05 in the CPRG abnormal cell proliferation analysis result in elevated levels of abnormal cell proliferation.

**Table 1 metabolites-13-00339-t001:** Number of tumor samples and control samples in each tumor type.

Tumor Type	Number of Tumor Samples	Number of Control Samples
BLCA	408	19
BRCA	1095	19
LUSC	501	51
LUAD	515	59
ESCA	184	13
LIHC	371	59
STAD	238	33
SKCM	466	5

**Table 2 metabolites-13-00339-t002:** Number of driver genes (positive labels) and nondriver genes (negative labels).

Tumor Type	Driver Genes	Nondriver Genes
BLCA	78	3586
BRCA	65	3646
LUSC	78	3586
LUAD	44	3690
ESCA	73	3659
LIHC	33	3636
STAD	35	3707
SKCM	14	3801

**Table 3 metabolites-13-00339-t003:** Parameters of GGraphSAGE framework.

Tumor Type	GAT (Input/Output Size)	GraphSAGE1 (Input/Output Size)	GraphSAGE2 (Input/Output Size)
BRCA	36/64	64/128	128/2
BLCA	36/64	64/128	128/2
ESCA	36/64	64/32	32/2
LUAD	36/256	256/64	64/2
LIHC	36/256	256/128	128/2
LUSC	36/128	128/32	32/2
STAD	36/128	128/64	64/2
SKCM	36/64	64/256	256/2

**Table 4 metabolites-13-00339-t004:** Number of samples of NPCD mutant/nonmutant group and median of abnormal cell proliferation level of sample groups.

Tumor Type	Gene Name	M/U Group Size	Median of Abnormal Cell Proliferation Level (M/U)
BRCA	CUL1	50/1045	0.971/0.551
	TBX3	85/1010	0.896/0.548
	MAP2K4	104/991	0.743/0.51
BLCA	ATP6V1G1	22/389	0.998/0.75
	ESF2	53/358	0.999/0.749
	ASXL2	64/347	1.12/0.561
ESCA	UTP3	23/161	1.25/0.987
	COL9A3	19/165	1.3/0.99
	PTCH1	32/152	1.131/0.748
LUAD	RIT1	86/481	0.801/0.562
	ATP6V0B	30/537	0.731/0.433
	MRPS12	30/537	0.799/0.561
LIHC	EZH2	79/292	0.688/0.49
	DYNC1H1	63/308	0.789/0.55
	NID2	48/323	0.727/0.501
LUSC	RILP	53/448	0.975/0.691
	EXOSC3	39/462	1.081/0.75
	KRR1	45/456	1.191/0.744
STAD	ERBB4	35/402	1.011/0.771
	ACTN2	48/389	1.232/0.822
	KIAA1429	28/156	1.088/0.752
SKCM	CR1	68/399	1.11/0.786
	PEG3	85/382	0.977/0.812
	EPHA3	48/413	1.134/0.78

## Data Availability

The data presented in this study are available at https://github.com/JP909/GGraphSAGE accessed on 3 July 2023.

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
