# Peer review of "Identification of Cancer Driver Genes by Integrating Multiomics Data with Graph Neural Networks"

_metabolites, 2023, doi:10.3390/metabo13030339_

Round 1

Reviewer 1 Report

In this paper, the authors propose a novel semi-supervised deep graph learning framework GGraphSAGE to predict cancer driver genes according to the impact of the alterations on the biological system. Different from most existing cancer driver identification methods which only consider genetic mutations, this paper has also considered non-genetic alterations and integrated multi-omics features for prediction. However, the authors need to address the following concerns before the MS could be considered for publication.

1.     Four categories of features including transcriptomic features, epigenomic features, genomic features, and network-derived features are used for prediction. The authors are suggested to evaluate the contribution significance of each type of feature.

2.     Regarding the experiment setting, the authors should give the parameter values of the GGraphSAGE framework in each case study.

3.     In line 306, 160 CPRGs are retrieved from GSEA. What pathways are included for these 160 CPRGs?

4.     In line 310, the authors select a set of the top 20 highly correlated genes. why is it set as 20?

5.     When evaluating the influence of mutated genes on cell proliferation, as shown in figure 7, boxplots do not reveal a significant difference between the two sets of values, but they do have strong statistical significance. Hence, authors are suggested to give the number of samples of a mutated group and non-mutated group for double check.

Reviewer 2 Report

The language of the manuscript should be checked again and text editing for a better understanding for the audience. 

Reviewer 3 Report

Excellent new algorithm to identify driver mutations, with a unique approach. Methods are well described and the metrics are pertinent. Overall a nice addition to the field. Once the major comment is addressed, this paper is publishable.

Major comment

Cancers with a higher mutation rate and more skewed signatures (e.g. UV) like melanoma - TCGA-SKCM should be analyzed using this method to test the robustness of the method. Most cancers studied in this paper have lower tumor mutational burden.

Minor comment

Figure 3 - typo gene expression

Round 2

Reviewer 3 Report

The authors have successfully answered all my comments and pointed out to other limitations in their revised manuscript. The data on SKCM is helpful and shows that this new algorithm can be applied to more mutated tumors.